# Bayesian inference is facilitated by modular neural networks with different time scales

**Kohei Ichikawa**[1]*, **Kunihiko Kaneko**[2,3]

**1** Department of Basic Science, Graduate School of Arts and Sciences, University of Tokyo, Meguro-ku, Tokyo, Japan, **2** Research Center for Complex Systems Biology, University of Tokyo, Bunkyo-ku, Tokyo, Japan, **3** The Niels Bohr Institute, University of Copenhagen, Blegdamsvej, Copenhagen, Denmark

* Kohei_Ichikawa@kk-generation.com

**Data Availability Statement:** Source codes for these models can be found at https://github.com/tripdancer0916/slow-reservoir.

**Funding:** This study was partially supported by a Grant-in-Aid for Scientific Research (A)

## Abstract

Various animals, including humans, have been suggested to perform Bayesian inferences to handle noisy, time-varying external information. In performing Bayesian inference by the brain, the prior distribution must be acquired and represented by sampling noisy external inputs. However, the mechanism by which neural activities represent such distributions has not yet been elucidated. Our findings reveal that networks with modular structures, composed of fast and slow modules, are adept at representing this prior distribution, enabling more accurate Bayesian inferences. Specifically, the modular network that consists of a main module connected with input and output layers and a sub-module with slower neural activity connected only with the main module outperformed networks with uniform time scales. Prior information was represented specifically by the slow sub-module, which could integrate observed signals over an appropriate period and represent input means and variances. Accordingly, the neural network could effectively predict the time-varying inputs. Furthermore, by training the time scales of neurons starting from networks with uniform time scales and without modular structure, the above slow-fast modular network structure and the division of roles in which prior knowledge is selectively represented in the slow sub-modules spontaneously emerged. These results explain how the prior distribution for Bayesian inference is represented in the brain, provide insight into the relevance of modular structure with time scale hierarchy to information processing, and elucidate the significance of brain areas with slower time scales.

## Author summary

Bayesian inference is essential for predicting noisy inputs in the environment and is suggested to be common in various animals, including humans. For the brain, to perform Bayesian inference, the prior distribution of the signal must be acquired and represented in the neural networks by sampling noisy inputs to estimate the posterior distribution of signals. By training recurrent neural networks to predict time-varying inputs, we demonstrated that those with modular structures, characterized by the main module exhibiting faster neural activity and the sub-module exhibiting slower neural activity, achieved highly

(20H00123) from the Ministry of Education, Culture, Sports, Science, and Technology (MEXT) of Japan and by MIC under a grant entitled "R&D of ICT Priority Technology (JPMI00316)" in part. KK is supported by Novo Nordisk Fonden. The funders had no role in study design, data collection and analysis, decision to publish, or preparation of the manuscript.

**Competing interests:** The authors have declared that no competing interests exist.

accurate Bayesian inference to perform the required task. In this network, the prior distribution was specifically represented by the slower sub-module, which effectively integrated the earlier inputs. Furthermore, this modular structure with different time scales and division of representing roles emerged spontaneously through the learning process of Bayesian inference. Our results demonstrate a general mechanism for encoding prior distributions and highlight the importance of the brain's modular structure with time scale differentiation for Bayesian information processing.

## Introduction

In the human and various animal brains, information processing involves inference based on inputs from the external world through the sensory systems, which obtain information from inputs under uncertainty [1] due to noise. To predict time-varying noisy inputs, previous studies suggested that animals such as humans and monkeys process inputs according to a Bayesian inference framework to deal with such uncertainty [2–13].

Bayesian inference is performed by calculating the posterior from the prior. The prior is gained from the history of inputs and gives the information to predict the signal input in advance. From it, the likelihood is estimated by observing the input signal. It is believed that prior knowledge must first be represented in the brain. It is then adjusted over time from the input history. However, how prior information is shaped in the brain remains elusive. In machine learning, several models such as variational recurrent neural networks (RNN) [14, 15] have been proposed that can perform Bayesian inference by computing prior from external signals. However, these models are designed specifically to make Bayesian inference, for instance, by introducing neurons for expressing prior, in advance. They, therefore, will not be able to answer how prior is shaped in the brain and which structure in the brain is relevant, in order to perform Bayesian inference. Here, we explore how neural networks acquire and represent prior knowledge, to predict time-varying noise inputs by performing Bayesian inference.

In this study, by recalling that a deterministic neural network with a simple learning algorithm can perform probabilistic inference [16], we investigate which type of RNNs can predict stochastic time-varying input better, by acquiring prior knowledge to perform Bayesian inference. Note that the acquisition mechanism of prior knowledge itself was studied earlier for fixed inputs [17]. In this case, however, fixed prior is sufficient, therein, and how time-varying prior was shaped in neural networks to predict time-varying inputs was not considered.

To discuss which structure of RNNs is relevant to shape the Bayesian inference, we recall hierarchical structure in the brain, with functional differentiated areas. In fact, some experiments suggest that the prior and the likelihood for Bayesian inference are encoded in different brain areas [18–20], even though the validity of the possible mechanism underlying the results remain controversial. On the other hand, relevance of area differentiation to the Bayesian inference can be theoretically expected as follows: In general, to obtain the prior, it is necessary to estimate the prior distribution based on previous observations. For it, the population of neurons representing the prior must integrate observed inputs over some time span. One possible mechanism for achieving such integration will be gained by adopting two neural modules functioning at distinct time scales; a downstream neuron population with slower activity changes separated from an upstream neuron population that processes input information. The existence of slow module that does not directly receive inputs in the neural network, thus will be relevant to integrate inputs over some time span.

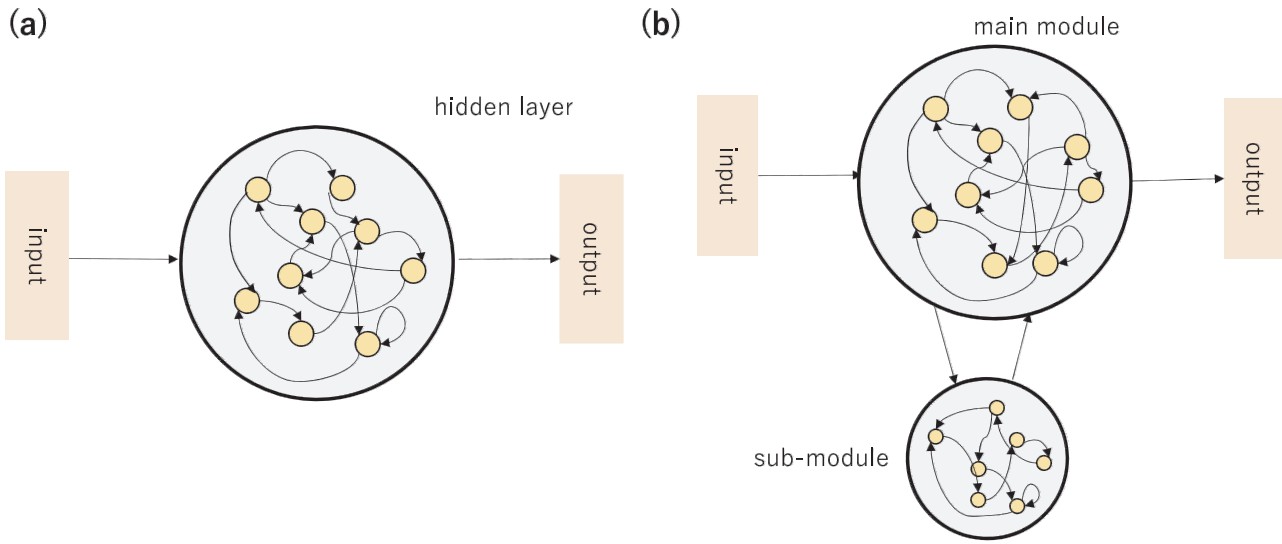

**Fig 1. Schematic of RNN.** (a) Standard RNN without modular structure (b) RNN with modular structure.

Some experimental reports, in fact, have suggested that the time scale of neural activities in higher layers of the brain that do not directly receive external input is slow [21–23], which may work to integrate the activities of lower layers. Note that putamen, amygdala, insula, and orbitofrontal cortex in the brain have been found to represent prior in an experiment [18].

Inspired by these experiments and theoretical considerations, we studied RNN models with two modules; a main module with a direct connection to the input-output layer and a sub-module with a direct connection to the main module and without connections to the input-output layer (i.e., a hierarchical structure)(Fig 1). In the RNN we applied a time-varying stochastic input, whose mean value changes with time whereas Gaussian noise is added around it. We trained the RNN to minimize the prediction error. Then, we confirmed that RNN could predict the input by the appropriate modular structure shaping the prior for the Bayesian inference. We examined a possible role of modular structure and the importance of the time scale difference between the main and sub-modules in forming the prior representation for Bayesian inference. We found that RNNs with a modular structure shape the prior more accurately than regular RNNs when the signal containing noise is inputted.

Further, Bayesian inference was shown to be more accurate when the time scale of the sub-module was appropriately slower. When the time scale was uniform, prior information was maintained in both the main module and sub-module. In this case, the performance of the prediction of time-varying input was rather low. In contrast, when the time scales were different, prior information was represented by the slow sub-module, in which case, the performance of prediction was quite high. In the latter case, the time variance of the prior was embedded in the neural manifold of the slower sub-module. With this embedding of variance by the sub-module, the average input change is clearly distinguished from noise, leading to better Bayesian inference.

In addition, we trained the RNN so that the connection of structure and time scale of neurons also change and examined if the modular structure with distinct time scales would emerge from a homogeneous neural network. As the training progressed, we observed that the time scales of neurons differentiated into slower and faster scales. A modular structure arose in which slower neurons were separated from the input/output layers, which were

predominantly connected to the fast neurons connecting input/output layers. This sub-module with slow neurons represented the prior information distinctly.

These results will be important to understand how the prior for Bayesian inference is represented in the neural networks and provide insight into the relationships between neural dynamics and the structure [24–28] underlying information processing in the brain.

## Materials and methods

### Recurrent neural networks with/without modular structure

To investigate the effect of structure and time scale on Bayesian inference, we considered the following RNNs [29]; a standard RNN without modular structure and RNN with a modular structure.

First, we adopted a standard RNN consisting of an input layer, a recurrent(hidden) layer, and an output layer, as shown in Fig 1(a). The number of neurons $N$ is set at $N = 200$ unless otherwise mentioned. The following equation represents the dynamics of the recurrent layer:

$$\mathbf{x}(t+1) = (\mathbf{I} - \boldsymbol{\alpha})\mathbf{x}(t) + \boldsymbol{\alpha}\text{ReLU}(W_{in}\mathbf{u}(t) + W\mathbf{x}(t)) + \sqrt{\boldsymbol{\alpha}}\boldsymbol{\xi}, \tag{1}$$

where $\mathbf{I}$ represents $\mathbf{I} = (1, 1, \ldots, 1)^{\mathsf{T}}$ and $\boldsymbol{\alpha} = (\alpha_1, \alpha_2, \ldots, \alpha_N)^{\mathsf{T}}$ represents a vector to introduce the rate of change, i.e. the inverse of time scale of the neurons($0 \leq \alpha_i \leq 1$) (Within this range of $\alpha$, the present model works). Furthermore, it should be noted that the products of vectors such as $(\mathbf{I} - \boldsymbol{\alpha})\mathbf{x}(t)$ are defined as the Hadamard product. If the timescale for neurons is identical, $\alpha_i$ is set as $\alpha_i = $ const. Below we consider the case with two timescales as

$$\alpha_i = \begin{cases} \alpha_m & (1 \leq i \leq N_m) \\ \alpha_s & (N_m < i \leq N), \end{cases} \tag{2}$$

where $N_m$ and $N - N_m$ neurons have distinct timescales. $N_m$ is set at $N_m = 150$ unless otherwise mentioned. Here the standard homogeneous network is given by $\alpha_s = \alpha_m$; Below, the case with $\alpha_s < \alpha_m$ was mostly studied to investigate the effect of time scale difference. Although we have mainly shown the results with a specific ratio of 150 fast neurons to 50 slow neurons, we found that the key results remained unchanged; In fact, we will show later that unless the number ratio is excessively biased (e.g., 10:240 or 240:10). As long as there are enough neurons in each category to capture the underlying dynamics, good Bayesian performance is achieved. (The condition may further depend on the details in the network complexity or specific interactions forms between fast and slow neurons, which will need further studies.) Here, $\mathbf{u}(t)$ is the input signal, and $\mathbf{x}$ is the state of the neurons in the recurrent layer. $W_{in}$ and $W$ represent the weight of synaptic connection. $\boldsymbol{\xi}$ was used to account for independent noise in dynamics given by a random variable that follows a normal distribution with mean 0 and standard deviation 0.05. We adopted the activation function ReLU(ReLU$(z) = 0$ for $z \leq 0$ and ReLU$(z) = z$ for $z > 0$) [30]. Then, the output of the RNN was determined by the linear combination of the internal states as follows.

$$\mathbf{y}(t) = W_{out}\mathbf{x}(t) \tag{3}$$

Next, we introduced a modular structure to the above RNN to ensure the distinction between the main and sub-modules(Fig 1(b)). Only the main module was connected to the

input/output layers. Thus, the dynamics of the recurrent layer are given by

$$\mathbf{x}_m(t+1) = (1 - \alpha_m)\mathbf{x}_m(t)+$$
$$\alpha_m \text{ReLU}(W_{in}\mathbf{u}(t) + W_{main}\mathbf{x}_m(t) + W_{s \to m}\mathbf{x}_s(t)) + \sqrt{\alpha_m}\xi_m \tag{4}$$

$$\mathbf{x}_s(t+1) = (1 - \alpha_s)\mathbf{x}_s(t)+$$
$$\alpha_s \text{ReLU}(W_{sub}\mathbf{x}_s(t) + W_{m \to s}\mathbf{x}_m(t)) + \sqrt{\alpha_s}\xi_s, \tag{5}$$

where $\mathbf{x}_m$ and $\mathbf{x}_s$ represent the firing rate of neurons in the main and sub-modules, respectively. Hence, $\alpha_m$ and $\alpha_s$ represent the inverse of time scale of the main and the sub-module, respectively. Here, $\alpha_m$ is fixed at 1(without losing generality), while we varied $\alpha_s$ from 1 to 0.01 to examine the effect of the time scale difference. The RNN output was determined by the linear combination of internal states of the main module.

$$\mathbf{y}(t) = W_{out}\mathbf{x}_m(t) \tag{6}$$

## Task

This study focused on a task optimally solved via Bayesian inference, in which an RNN was assigned to estimate the actual value from a noise-perturbed signal. We constructed the external input signal $s$ as follows:

1. Initially, we randomly sampled the true value, $y_{true}$, from a generator (or cause) distribution defined by a normal distribution with a mean of $\mu_g$ and a variance of $\sigma_g^2 : y_{true} \sim \mathcal{N}(\mu_g, \sigma_g^2)$

2. Subsequently, the $s$ was generated from $y_{true}$ by adding noise which is sampled from the normal distribution with mean 0 and variance $\sigma_l^2 : s \sim \mathcal{N}(y_{true}, \sigma_l^2)$

Here it is important to note that the generator is not static, and is changed over time with a probability $p_t$, where $p_t$ represents the parameter indicating the likelihood of changes in the generator. Upon the change, the parameters $\mu_g$ and $\sigma_g$ were updated to values uniformly sampled from a given ranges: $\mu_g \in [-0.5, 0.5]$ and $\sigma_g \in [0, 0.8]$. When the generator model alters, it changes the distribution to which $y_{true}$ conforms. As a result, the external input signal $s$ fed into the RNN also changes. However, even if the generator changes, the signals before and after the change are combined and input to the RNN as a series of signals.

As mentioned in the Introduction, we set this task, as the RNN can make predictions when it performs Bayesian inference. To perform it, the prior distribution needed for Bayesian inference must be estimated from the observed signal so that it is close to the generator distribution. $\mathbf{u}(t)$ for Eq 1 (or 4 and 5) is given by the Probabilistic Population Code (PPC) [31]. Now we need to assign the input term $\mathbf{u}(t)$ in Eq 1 from the signal $s$. For it, we adopted the Probabilistic Population Code (PPC) [31]. PPC assumes that the information in a signal is encoded by a population of neurons with a position-based preferred stimulus that fires probabilistically according to a Poisson distribution. It has been shown that neural networks with a population of neurons with input by PPC can learn probabilistic inference effectively [16]. Therefore, in this study, we also assumed that the activity $\mathbf{u}$ of the input-layer neurons encoding the observed signal followed the PPC model. Accordingly, $\mathbf{u}$ was sampled from the following

Poisson distribution [32] every time step:

$$p(\mathbf{u}|s) = \prod_i \frac{e^{-f_i(s)} f_i(s)^{u_i}}{u_i!} \tag{7}$$

Here, $s$ is the observed input signal (i.e., which is generated from $y_{true}$ by adding noise), whereas $f_i$ is the tuning curve of the neurons, which represents how responsive each neuron is to $s$. The signal variance $\sigma_l^2$ is inversely encoded in $f_i$ as the amplitude of the tuning curve. The tuning curve of the neurons is represented as follows:

$$f_i(s) = \frac{1}{\sigma_l^2} \exp\left(\frac{-(s - \phi_i)^2}{2\sigma_{\mathrm{PPC}}^2}\right), \tag{8}$$

where $\phi_i$ represents the preferred stimuli of neurons in the input layer. It was assumed that $\phi_i$ follows an arithmetic sequence for $i$ ($\phi_i = -1/2 + i/m$ when the number of neurons in the input layer is $m$) [33]. Also, $\sigma_{\mathrm{PPC}}^2$ is a constant that represents the width of tuning curve which was set as $\sigma_{\mathrm{PPC}}^2 = 1/2$ in this study. By employing the above of tuning curve transformation, it is demonstrated that one can encode the information from the external signal $s$ into the spatial position of neurons that are most likely to fire.

In this task, the true value $y_{true}$ was to be estimated based on the input signal $\mathbf{u}$. Therefore, training was performed to minimize the mean squared error (MSE) between the neural network output $y(t)$ and the true value $y_{true}(t)$. Note that the information of $y_{true}$ is used only to calculate the loss function when learning. (We acknowledge that providing the true value $y_{true}(t)$ for training might be an artificial setting. However, similar settings in which the true value is provided have been utilized in previous studies, such as in the work by [18]. For the purposes of this study, to investigate the role of a modular structure and time-scale difference, it will be useful to adopt the simple and previously established settings. Then one can compare the present system with the standard homogeneous network case, even though it cannot fully reflect on the real-world situation.)

$$L = \frac{1}{T} \sum_t \left(y(t) - y_{true}(t)\right)^2 \tag{9}$$

The training was performed by using the backpropagation method [34, 35], to decrease $L$ by optimizing the weight of synaptic connections $W_*$. Here, this optimization is performed by the stochastic gradient method. For it, an efficient method called Adam [36] is generally established and widely use, which was adopted here. The batch size of training samples was set to 50. In machine learning, a batch size refers to the number of training samples that are processed together in one iteration. The weight decay rate was set to 0.0001, where the weight decay is a regularization technique used in neural networks to prevent overfitting. This decay is introduced by adding a penalty, proportional to the size of the weight coefficients, to the loss function that the model is asked to minimize. By setting a weight decay rate, we ensure that the model does not too much focus on some particular feature and can generalize better to unseen data. In each iteration in machine learning, the network performs a single pass through the entire training dataset. Here we set 6000 iterations for the training. Here, in our case, the training process was performed over the complete set of training data 6000 times. This is a typical number adopted in machine learning and we also confirmed that this number is sufficient to complete the training. See Table 1 for the hyperparameters used in the experiment.

**Table 1. Hyperparameters.** Important items are in bold.

| Attribute | Value |
|---|---|
| Range of $\mu_g$ | $-0.5 \leq \mu_g \leq 0.5$ |
| Range of $\sigma_g$ | $0 \leq \sigma_g \leq 0.8$ |
| Range of $\sigma_l$ | $\sqrt{1/5} \leq \sigma_l \leq 1$ |
| **Switching probability of the generator**: $p_t$ | $\mathbf{p_t = 0.03}$ |
| Dimension of the input units $\mathbf{u}(t)$ | 100 |
| $\sigma_{\mathrm{PPC}}$ | 0.5 |
| Lasting time of $\mathbf{u(t)}$ | $T = 120$ |
| **# of neurons in the main module** | $\mathbf{N_m = 150}$ |
| **# of neurons in the submodule** | **50** |
| $\boldsymbol{\alpha}_m$ | **1** |
| $\boldsymbol{\alpha}_s$ | **1, 0.5, 0.2, 0.1, 0.05, 0.01** |
| Batch size | 50 |
| Optimization algorithm | Adam |
| Learning rate | 0.001 |
| Iteration | 6000 |
| Weight decay | 0.0001 |

## Results

### Fixed structure and time scales

**Bayesian optimality.** Because the generated signal $s$ was observed under noise, the neural network was required to estimate the true value $y_{true}$ sampled from the generator. If the information from the generator were known, we can estimate the true value optimally as follows (maximum a posteriori(MAP) estimation [37]).

$$y_{opt} = \frac{\sigma_g^2}{\sigma_g^2 + \sigma_l^2} s + \frac{\sigma_l^2}{\sigma_g^2 + \sigma_l^2} \mu_g \tag{10}$$

However, as described in the "Task" section, the information from the generator was not explicitly given to the neural network, so it must be estimated from observed signals as a prior distribution. First, we examined whether the neural network could achieve this prior-based estimation.

The output $y$ of RNN with modular structure trained with $\alpha_s = 0.1$, when given an observed signal $s$, is shown Fig 2. $s$ was sampled from the prior with $\mu_g = 0.5$, $\sigma_g = 0.5$, and $\sigma_l = \sqrt{1/5}$ of noise was added. The green points represent the estimation based on the maximum likelihood estimation $y_{ML}$, which is with the highest accuracy when no prior information is available. Here, this estimation is nothing but matching with the observed signal $s$. The blue points represent $y_{opt}$ when estimated according to the MAP estimation, and the orange points represent the actual neural network output $y$. Fig 2 shows that the output of RNN is closer to the blue points $y_{opt}$ rather than to the green points, indicating that approximate (nearly-optimal) Bayesian inference with a well-estimated prior is achieved (the mean squared error between $y$ and $y_{ML}$ is 0.15, and the mean squared error between $y$ and $y_{opt}$ is 0.019, the latter being smaller).

Next, we examined the optimality of the Bayesian estimation for networks with and without modular structures and time scale differences. Fig 3(a) shows the MSE between $y$ and $y_{opt}$ by the RNN trained under each condition. This result shows that the modular structure improved

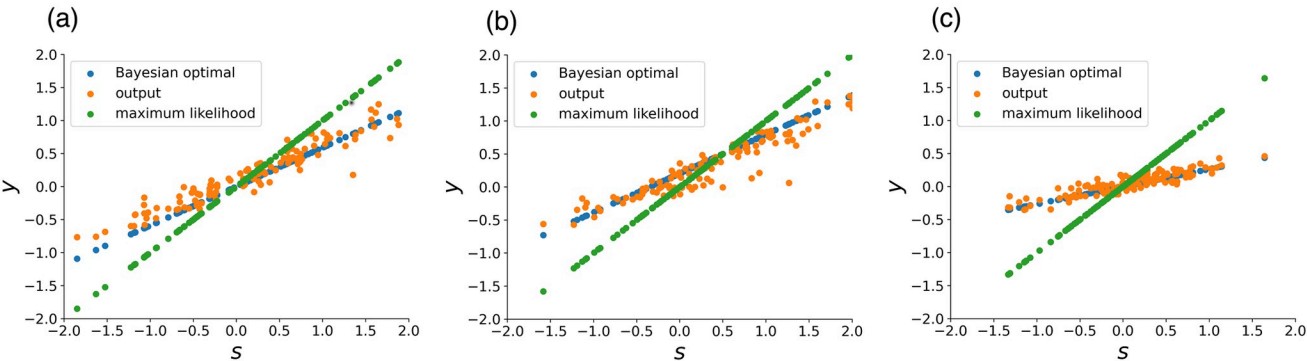

**Fig 2.** (a) The output $y$ of RNN against the observed signal value $s$. Before $s$ is input, the time series signal, which is sampled from the normal distribution with the mean $\mu_g = -0.3$ and the standard deviation $\sigma_g = 0.3$ and then the noise with the standard deviation $\sigma_l = \sqrt{1/5}$ is added in the input. The accuracy can be increased by estimating prior based on the signal input before $s$ and performing Bayesian inference. Blue points represent $y_{opt} = \frac{\sigma_g^2}{\sigma_g^2 + \sigma_l^2} s + \frac{\sigma_l^2}{\sigma_g^2 + \sigma_l^2} \mu_g$ value, orange points represent the output of RNN $y$, and green points represent estimation based on maximum likelihood estimation $y_{ML} = s$. The result is for a model with $\alpha_s = 0.1$. (b) The same plots as (a), with the settings $(\mu_g, \sigma_g) = (0.3, 0.3)$. When compared to (a), the output $y$ of the RNN shows a larger value as expected by the Bayesian optimal. (c) The plots for the settings $(\mu_g, \sigma_g) = (-0.3, 0.6)$. Compared to (a), the output $y$ of the RNN shows a lower slope as expected by the Bayesian optimal.

the accuracy of Bayesian estimation, which was further increased when $\alpha_s$ decreased to an appropriate degree. In fact, we found the optimal time scale $\alpha_s = 0.06 \sim 0.2$, at which the maximum accuracy was achieved. As shown in S4 Fig MSE remains to be low at around $0.06 \lesssim \alpha_s \lesssim 0.2$. Even without modular structure, the time scale difference contributed to inference accuracy, but the accuracy increased significantly with both the modular structure and time scale difference.

**Adjustability to rapid generator switching.** So far, we studied the performance of Bayesian inference models under a fixed generator to compare the accuracy of Bayesian inference

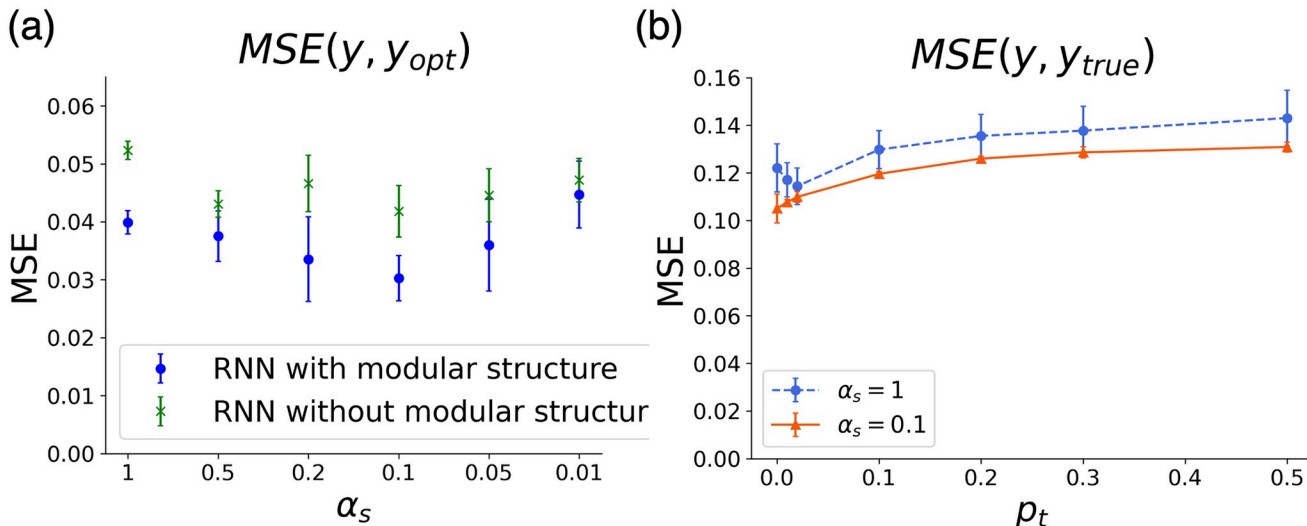

**Fig 3.** (a) MSE between the optimal value $y_{opt}(t)$ and the output of RNN $y(t)$, plotted against the time scale $\alpha_s$. · with modular and × without modular structure. RNNs with a modular structure are more accurate. In addition, those with $\alpha_s \sim (0.1 \sim 0.2)$ have optimal error. (b) MSE between the true value $y_{true}(t)$ and the output of RNN $y(t)$ for the network with $\alpha_s = 1(·)$ and $\alpha_s = 0.1(×)$. The value increases as $p_t$ increases, but the model with $\alpha_s = 0.1$ is always more accurate.

itself. Next, we examined their performance when the generator changes in time. To perform Bayesian inference for a rapidly changing input, it was necessary for the model to quickly approach the new optimal value $y_{opt}$ to yield a good estimation. To verify the accuracy of the RNN in this case, we compared the MSE between $y_{true}(t)$ generated by the generator and the output $y(t)$ of RNN under various $p_t$(Fig 3(b)). The model with $\alpha_s = 0.1$ was found to be more accurate for all values of $p_t$.

As a special case, we considered a setting where the input moves back and forth between two generators, A and B. Then we examined whether the prior distribution estimated by the RNN was closer to the distribution of either generator. Specifically, we adopted generator A with $(\mu_g, \sigma_g^2) = (\mu_A, \sigma_A^2)$ and generator B with $(\mu_g, \sigma_g^2) = (\mu_B, \sigma_B^2)$ and compute the following values when the Bayesian optimal estimates under each generator were $y_{opt}^A, y_{opt}^B$.

$$a(t) = \frac{y_{opt}^B - y(t)}{y_{opt}^B - y_{opt}^A} \tag{11}$$

When $a(t)$ is close to 1, the model's prior is closer to generator A, and when $a(t)$ is close to 0, it is closer to generator B.

Comparing the change in $a(t)$ between the model with $\alpha_s = 0.1$ and the model with $\alpha_s = 1$, we found that the model with $\alpha_s = 0.1$ was more adjustable to the generator change, as shown in Fig 4(a). This result shows that the model with $\alpha_s = 0.1$ was more responsive to the changes of the generators and recognized the generator change more quickly in all runs. The difference between the two models was especially pronounced in the extreme case in which the two generators switched every time(Fig 4(b)). Intuitively, having a population of slow neurons would seem to be disadvantageous in responding to rapid environmental changes, but the results showed the opposite. The network with $\alpha_s = 1$ could not follow rapid input changes, whereas that with $\alpha_s = 0.1$ could estimate the input prior effectively. We discuss the importance of slow neurons in responding to rapid changes below. Furthermore, we also checked that when $\mu_g$

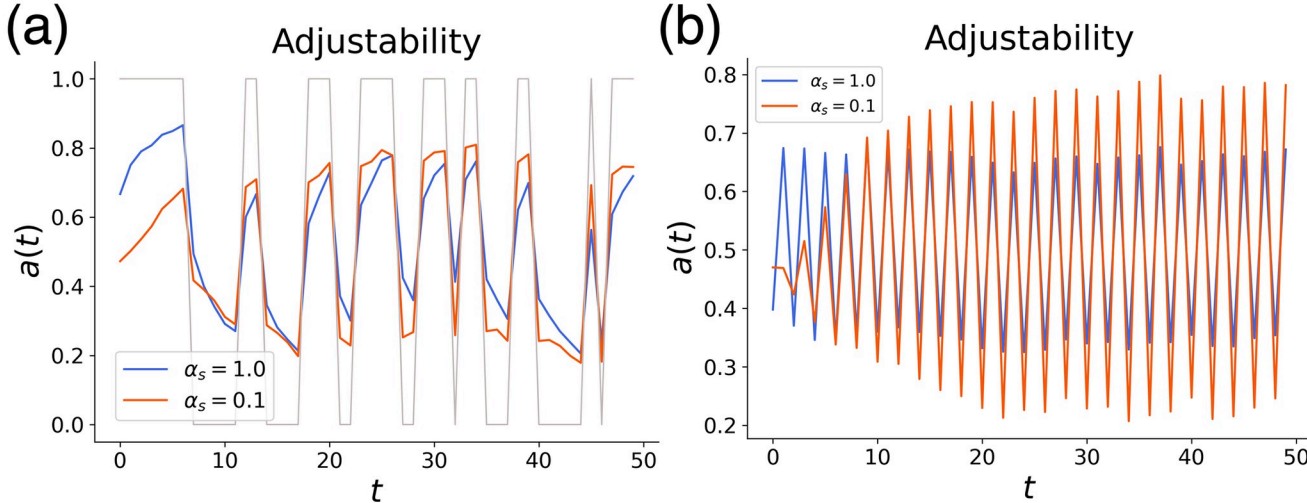

**Fig 4. Adjustability to rapid generator change.** (a) $a(t)$ for the case where generator A($(\mu_A, \sigma_A^2) = (-0.5, 0.04)$) and generator B($(\mu_A, \sigma_A^2) = (0.5, 0.04)$) switch alternately with probability $p_t = 0.2$. The model with $\alpha_s = 0.1$ adjusted more quickly to the generator change. The thin line represents $a(t)$ when the output was fully adjusted to generator switching. (b) $a(t)$ for the case where generator A($(\mu_A, \sigma_A^2) = (-0.5, 0.04)$) and generator B($(\mu_A, \sigma_A^2) = (0.5, 0.04)$) switch every time(periodic switch).

and $\sigma_g$ are constant, the modularity of the network is not necessary, and the difference in the performance with and without modularity was not detected.

**Representation of the prior.** We investigated how the slow sub-module facilitated improved prior representation for Bayesian inference. By starting from the examination of the hypothesis that a group of downstream slow neurons represent the prior by integrating the observed signal over time, we investigated which side of the main/sub-module was responsible for the prior information in the modular RNN.

Here, by using the prior information, the estimated value was shifted from the observed signal $s$ to an appropriate value $y_{opt}$(Eq 10). In other words, even given the same signal input $s$, the output varied depending on which time series signal was input before $s$ (because the prior estimation changed). Even if one module returned to its original state, the output shifted from $s$ because the prior information remained in the other module. The magnitude of this change is considered to represent the degree to which the module utilizes the estimated prior information. Therefore, it is possible to estimate the extent to which each module plays a role in prior information processing by examining the change in the output $y(t)$ when the internal state of each main and sub-module is changed to the value corresponding to a different prior.

First, let $\mathbf{x}_m(t; \mu_g, \sigma_g)$, $\mathbf{x}_s(t; \mu_g, \sigma_g)$ be the internal states of the main and sub-module, respectively, when the input signal $s$ from a generator $(\mu_g, \sigma_g)$ is applied for a certain period. Because the output $y(t)$ is determined by the internal states of two modules and the input signal at $t-1$, it can be written as $y(\mathbf{x}_m(t-1; \mu_g, \sigma_g), \mathbf{x}_s(t-1; \mu_g, \sigma_g), s(t-1), \sigma_l)$(From now on, time notation will be omitted). From this, the change in the output $y$ is computed by fixing one of the two modules and varying the other to a different internal state $\mathbf{x}_i(\mu_g, \sigma_g) \rightarrow \mathbf{x}_i(\mu'_g, \sigma'_g)$. This change is made by saving the value of internal state $\mathbf{x}_{m,s}$ obtained by applying the input $s$ created from the generator of $(\mu'_g, \sigma'_g)$, and by changing the value of $\mathbf{x}_{m,s}$ to that value during RNN inference. (This procedure is adopted only for the sake to analyze which module is more responsible for the prior information.) The degree of change in $y$ represents the impact on the output of each module reflecting the prior information. Hence, by comparing the above variances of $y$ by $\mathbf{x}_m$(or $\mathbf{x}_s$) with fixed $\mathbf{x}_s$(or $\mathbf{x}_m$) respectively, it is possible to estimate how much each module is responsible for the prior representation. Specifically, we fixed one of the modules at the values with $\mu_g = 0$ and $\sigma_g = 0.4$ (These values are set to the median of the range of values $-0.5 \le \mu_g \le 0.5$ and $0 \le \sigma_g \le 0.8$), i.e., $\mathbf{x}_i(0, 0.4)$, while for the other module, $\mu_g$ and $\sigma_g$ are changed as $\mathbf{x}_g(\mu_g, \sigma_g)$. Then, we calculated the variance of $y$ as

$$V_m = \langle \mathrm{Var}[y(\mathbf{x}_m(\mu_g, \sigma_g), \mathbf{x}_s(0, 0.4), s, \sigma_l)]_{(\mu_g, \sigma_g)} \rangle_{(s, \sigma_l)} \tag{12}$$

$$V_s = \langle \mathrm{Var}[y(\mathbf{x}_m(0, 0.4), \mathbf{x}_s(\mu_g, \sigma_g), s, \sigma_l)]_{(\mu_g, \sigma_g)} \rangle_{(s, \sigma_l)}, \tag{13}$$

where $\mathrm{Var}[\ ]_{(\mu_g, \sigma_g)}$ denotes the variance over the changes of $(\mu_g, \sigma_g)$, and $\langle \rangle_{(s, \sigma_l)}$ denotes the average over the changes of $(s, \sigma_l)$. The magnitudes of $V_s$ and $V_m$ indicate the extent to which the sub-module and main module, respectively, influence the variation in the output, to respond upon changes in the signal's prior distribution.

Dependencies of $V_s$ and $V_m$ on different $\alpha_s$ are shown in Fig 5. This result shows that when $\alpha_s = 1$ (i.e., the time scale is uniform), both the main and sub-modules contribute to the representation of prior distribution to the same degree. Conversely, when $\alpha_s = 0.05 \sim 0.5$, $V_s$ is much larger than $V_m$, meaning that the sub-module selectively contributes to the representation of the prior. In particular, when $\alpha_s = 0.1$ and $0.2$, the differentiation of representation between the main and sub-modules is more pronounced. Note that the contribution of the main module is large when $\alpha_s = 0.01$, probably because the time scale of the sub-module is too

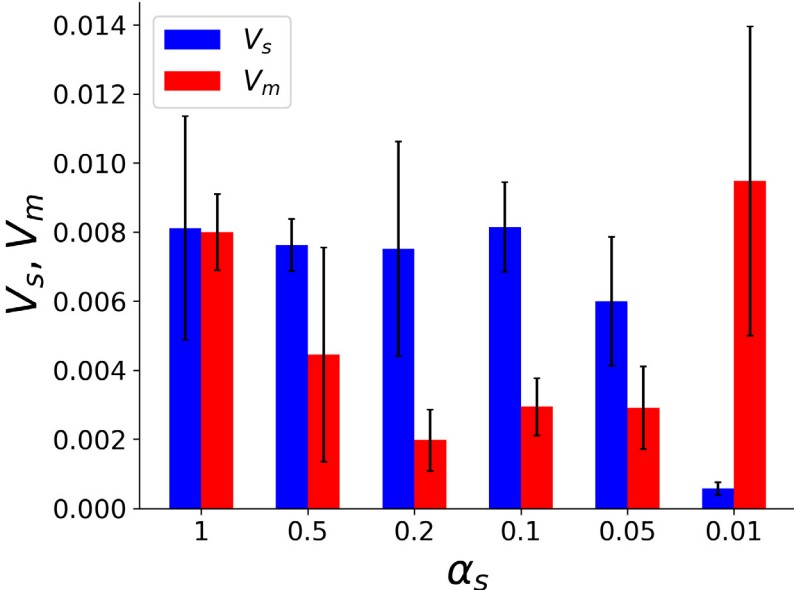

**Fig 5. Division of roles for representing prior distribution.** $V_s$, $V_m$ defined in the text Eqs (12) and (13) plotted for different values of $\alpha_s$ computed over 1000 samples of data. $V_s$ and $V_m$ represent the degree to which the sub-module and the main module are responsible for prior-based information processing. When $\alpha_s = 0.05 \sim 0.5$, in particular for $\alpha_s = 0.1$ and 0.2, the sub-module selectively contributes to the representation of the prior.

slow to code the information of the prior. Comparing Figs 3 and 5 shows that the highly accurate Bayesian inference is achieved when the prior distribution information is localized in the sub-module.

Next, we investigated how the prior is represented by the main and sub-modules by visualizing the neural activity by principal component analysis(PCA) [38, 39]. First, $\mathbf{x}_m(\mu_g, \sigma_g)$ and $\mathbf{x}_s(\mu_g, \sigma_g)$ were computed for various $(\mu_g, \sigma_g)$ in a model with $\alpha_s = 0.1$, and made PCA. The results were projected on a plane using the first and second principal components and color-coded according to $\mu_g$ and $\sigma_g$ (Fig 6(a) and 6(b)). The neural activity in the main module was loosely distributed on a one-dimensional manifold, represented by the first principal component(PC1). This PC1 approximately corresponded to the $\mu_g$ value, although the distinction was not clear. In contrast, the activity in the sub-module was clearly represented by 2-dimensional manifolds, as in Fig 6(b2), where PC1 corresponds to $\mu_g$, and PC2 corresponds to $\sigma_g$, rather well.

Then, we performed the same analysis on the model with $\alpha_s = 1$ (Fig 6(c) and 6(d)). In this case, the manifolds of neural activities for the main and sub-modules did not change significantly. Both were represented in a one-dimensional manifold corresponding to $\mu_g$; there was no axis corresponding to $\sigma_g$. The decodability of $\sigma_g$ achieved in the internal states of the sub-module with $\alpha_s = 0.1$ was not observed for $\alpha_s = 1$. In fact, the coefficient of determination when $\sigma_g$ was calculated by Ridge regression from the internal state of the sub-module with $\alpha_s = 0.1$ was 0.68, while that using the sub-module with $\alpha_s = 1$ is −0.03. This suggests that the model with $\alpha_s \sim 0.1$ can better distinguish the input's variance from noise to perform Bayesian inference accurately.

When the generator changed rapidly, the variance of the prior was larger than the variance of the generator, as shown in the S1 Fig for the case with $\alpha_s = 0.1$. When $\sigma_g$ was large, as seen from Eq 10, the influence of the observed signal $s$ was larger than that of $\mu_g$, allowing the model

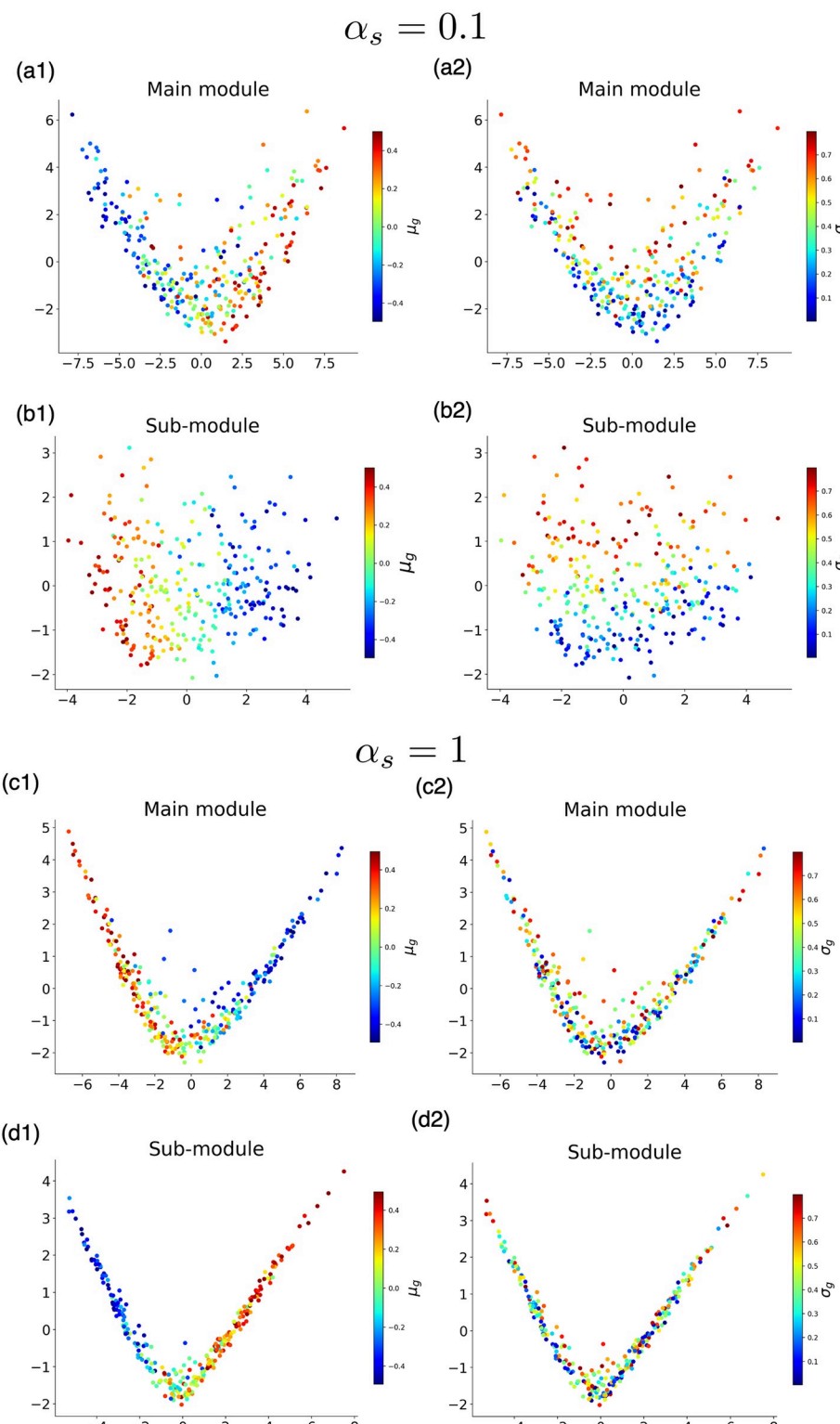

**Fig 6. The neural activities of the main module $x_m(\mu_g, \sigma_g)$ and sub-module $x_s(\mu_g, \sigma_g)$ were plotted by the first and second principal component spaces.** (a,b) is the result of $\alpha = 0.1$, and (c,d) is of $\alpha_s = 1$. (a1,c1) Main module, color-coded by $\mu_g$. (a2, c2) Main module, color-coded by $\sigma_g$. (b1,d1) Sub-module, color-coded by $\mu_g$. (b2, d2) Sub-module, color-coded by $\sigma_g$. 300 data are plotted.

to "keep up" with large changes in the observed signal. This explains the higher adjustability to rapid generator changes as seen in Fig 4.

To investigate whether the division of roles and accurate Bayesian inference depend on the number of neurons in the sub-module and main-module, we trained models with varying the ratio values of $N_s$ to $N_m$ by fixing at $N_s + N_m = 250$. As shown in S3 Fig, the MSE remained to be low as long as the number fraction is not too biased(e.g., 10:240 or 240:10). Except these extreme cases, the efficient Bayesian inference was achieved, where the division of roles was achived as computed by $V_s$ and $V_m$. Here, to investigate whether the difference in time scale or the existence of slower time scale itself was more influential, we trained an RNN with $\alpha_s = \alpha_m = 0.1$ and examined its accuracy. As shown in S2 Fig, We found that when $(\alpha_m, \alpha_s) = (0.1, 0.1)$, the MSE was larger, and the accuracy was worse than that in the cases with $(\alpha_m, \alpha_s) = (1, 1)$ and $(\alpha_m, \alpha_s) = (1, 0.1)$. Therefore, it is not simply the slower time scale of the neurons but the time scale difference between the main and sub-modules facilitates the accuracy in Bayesian inference.

**Effects of different time scales.**  To examine the impact of $\alpha_s$ differences on Bayesian inference accuracy in detail, we considered how each model with $\alpha_s = 1$ and $\alpha_s = 0.1$ represents prior as a function of the input signal $s(t)$.

The RNN need to estimate the current generator from past input signals in order to accurately predict $y(t)$. In this paper, this estimation is treated as a prior. Thus, we assume that the mean $\mu_p$ in the current generator estimated by the RNN is represented as a superposition of past input signals as follows:

$$\mu_p \simeq \sum_k a_k s(t - k).$$

(14)

Note that $\mu_p$ corresponds to the "current state of the generator" estimated by the RNN and is treated as a variable distinct from $\mu_g$ (for example, right after the generator switches from $\mu_g = -0.5$ to $\mu_g = 0.5$, if a positive $s(t)$ is input into the RNN, the RNN would estimate that the mean of generator is remains to be still $-0.5$). If the values of $a_k$ in the above equation are known, it will be possible to discuss how the RNN estimates the current state of the generator, and how it performs this estimation using the past input signals $s(t)$.

Below, we estimate $a_k$. For it, it is necessary to estimate $\mu_p$. Here, there is the one-to-one relationship between the internal state **x** of the RNN and $\mu_p$. Given this background, we define $\mu_p$ as $\mu_p = W_{\mu_p} \mathbf{x}$. Here, $W_{mu_p}$ is considered to be a transformation matrix, calculated as follows: We fixed the generator and estimated $W_{\mu_p}$ by calculating **x**. Then, we created a data vector $M_g$ that gives the time series $\mu_g^1, \mu_g^2, \ldots$ and a data matrix $X$ that gives the time series of the internal states $\mathbf{x^1}, \mathbf{x^2}, \ldots$ i.e.,

$$M_g = (\mu_g^1, \mu_g^2, \ldots)^\mathsf{T}, \ \ X = (\mathbf{x^1}, \mathbf{x^2}, \ldots)$$

Then, we seeked for the matrix $W_{\mu_p}$ such that $M_g \simeq W_{\mu_p} X$. Using the Moore-Penrose pseudo inverse, we obtain the best-fit matrix as $W_{\mu_p} = M_g X^\dagger$ [40]. Let $\mu_p$ be the result of the transformation by $W_{\mu_p}$. As Fig 7(a) shows, $\mu_g \simeq \mu_p$ is valid.

Based on the calculation of $\mu_p$, we estimated $a_k$ by the following steps. First, we obtained **x**(t) against the time-varying signal with a probability of $p_t = 0.03$. By applying the above transformation matrix, $W_{\mu_p}$ to $\mathbf{x}(t)$, which was obtained at this time, the prior $\mu_p$ was estimated. The state of the prior was thus obtained for the time series of the observed signal $s(t)$.

Finally, $a_k$ in Eq 14 was obtained by minimizing the difference between the two sides of Eq 14. Specifically, we created a data vector $M_p$ that arranges $\mu_p$ and a data matrix $S$ that arranges

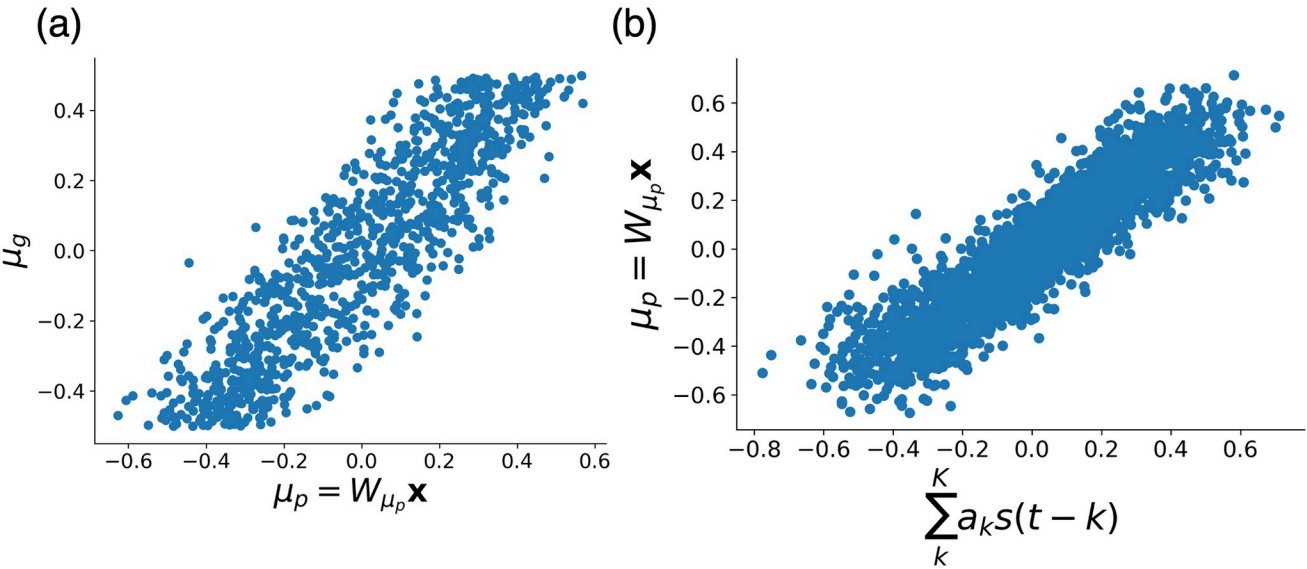

**Fig 7.** (a)Comparison between the estimated mean of prior $\mu_p$ and the mean of generator $\mu_g$. (b)Comparison between the linear weighted sum of past signals $s(t-k)$ and the estimated mean of prior $\mu_p$.

$\mathbf{s} = (s(t-1), s(t-2), \ldots, s(t-K))^\mathsf{T}$. Using the Moore-Penrose pseudo inverse, we obtained $\mathbf{a} = (a_1, a_2, \ldots, a_K)^\mathsf{T}$. As Fig 7 shows, $\mu_p \simeq \sum_k^K a_k s(t-k)$ was valid.

Because the obtained coefficients correspond to the contribution of the signal before $k$ time steps, we could estimate the extent to which the neural network uses past information in estimating the prior.

The estimated coefficients of Eq 14 were plotted against $k$(Fig 8), revealing that the model with $\alpha_s = 0.1$ used more past information in estimating prior information than the model with $\alpha_s = 1$. This difference in time windows leads to a difference in accuracy for prior encoding.

## Organization of modular structure time-scale separation

So far, we have investigated neural networks with fixed and modular structures along fixed time scales and demonstrated that those with fast and slow modules effectively represented the prior distribution. Then, we investigated whether such a modular structure would emerge by training a neural network to predict $y_{true}$ from a homogeneous-structure network. Here, it should be noted that our findings about the effectiveness of modular structures with slow/fast time scales in Bayesian inference do not necessarily imply such structure emerges through learning. In this section, we examine if slow/fast separation with corresponding modular structure is reachable just by learning.

We again used the same neural network model as Eq 1. In this section, $\boldsymbol{\alpha}$ values, as well as elements of $W$, change by training to start from initial values set randomly according to $\mathcal{N}(0.5, 0.1)$. In other words, we examine if the modular structure together with the timescale difference emerges from the random Gaussian distribution without such structure. During training, each matrix $W$ and $\boldsymbol{\alpha}$ are optimized according to the gradient descent method [41] at each step. The number of neurons in the recurrent layer of the neural network was set to 80.

The change in $\boldsymbol{\alpha}$ distribution during the learning task is shown in Fig 9(a). As shown, $\boldsymbol{\alpha}$ split into two groups over the learning period: one with large values close to 1 and the other with small values near 0.1.

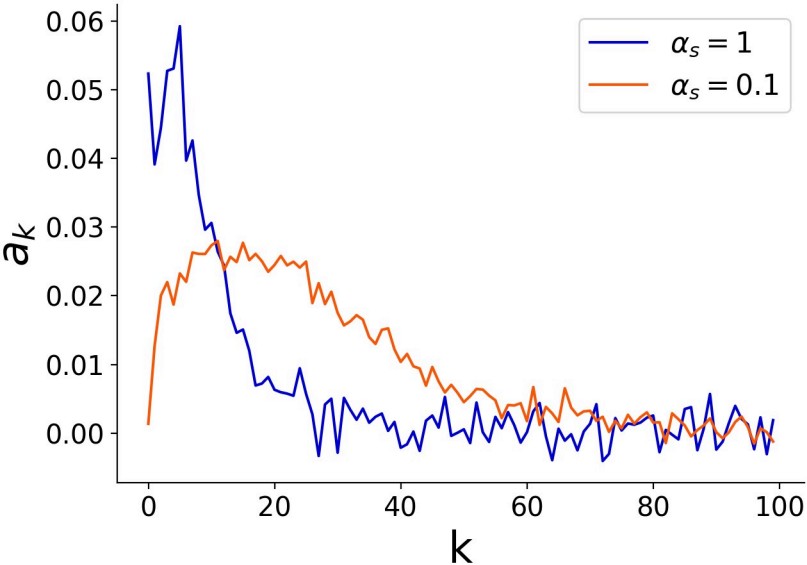

**Fig 8. $a_k$ defined by Eq 14 is plotted against $k$, for the model with $\alpha_s = 1$ and $\alpha_s = 0.1$ using 3000 data points.** Note that $a_k$ corresponds to the coefficient when the mean of the current generator that the RNN is estimating is expressed as a superposition of past signals $s(t)$, given by $\mu_p \simeq \sum_k a_k s(t - k)$. From this, it can be seen that the model with $\alpha_s = 0.1$ used more past information in estimating prior information than the model with $\alpha_s = 1$.

Next, we measured the contribution of prior representation as examined in the "Representation of the prior" section for groups of neurons with large values $\boldsymbol{\alpha}$ (neurons with $\alpha_i > 0.8$) and groups of neurons with small values $\boldsymbol{\alpha}$ (neurons with $\alpha_i < 0.2$) for three epochs in the learning process(Fig 9(b)). We found that after 10000 epochs, the slow neurons were responsible for the representation of prior distribution, as in the model with $\alpha_s = 0.1$ in the fixed time scale setting. To explore the optimal time scale $\boldsymbol{\alpha}$ in response to the variations in $p_t$, we allowed $\boldsymbol{\alpha}$ to be trainable, by taking an approach similar to that in the last chapter. Models were trained under three distinct settings: $p_t = 0.03, 0.1,$ and $0.3$. The results are illustrated in S5 Fig. As shown, the peak on the smaller $\boldsymbol{\alpha}$ side shifted to a larger value as $p_t$ increased. Conversely, the peak on the larger $\alpha$ remained constant at 1 across all settings, with no discernible difference in its proportion.

Finally, we investigated the neural network structure shaped by training. In Fig 9(a), the recurrent layer neurons of the network of epoch 10000 were split into the three groups, divided by the magnitude of $\alpha_i$, slow neurons with $\alpha_i < 0.2$, fast neurons with $\alpha_i > 0.8$, and $0.2 \leq \alpha_i \leq 0.8$ neurons as the others. The average connectivity between the input layer, each group, and the output layer is shown in Fig 9(c) [42]. The connection from the input layer to the group of fast neurons and that from the fast neurons to the output layer were distinctively larger than those to or from the slow neurons. Among connections within the recurrent layer, those between the fast and slow neurons were larger than others. In summary, a modular structure, shown in Fig 1(b), emerged through learning alone.

## Discussion

In this study, we demonstrated that neural networks with slow and fast activity modules play an essential role in the prior representation for Bayesian inference. We set up a task to predict a time-varying signal under noise that could be estimated by Bayesian inference and trained RNNs with or without modular structure and with or without time scale differences.

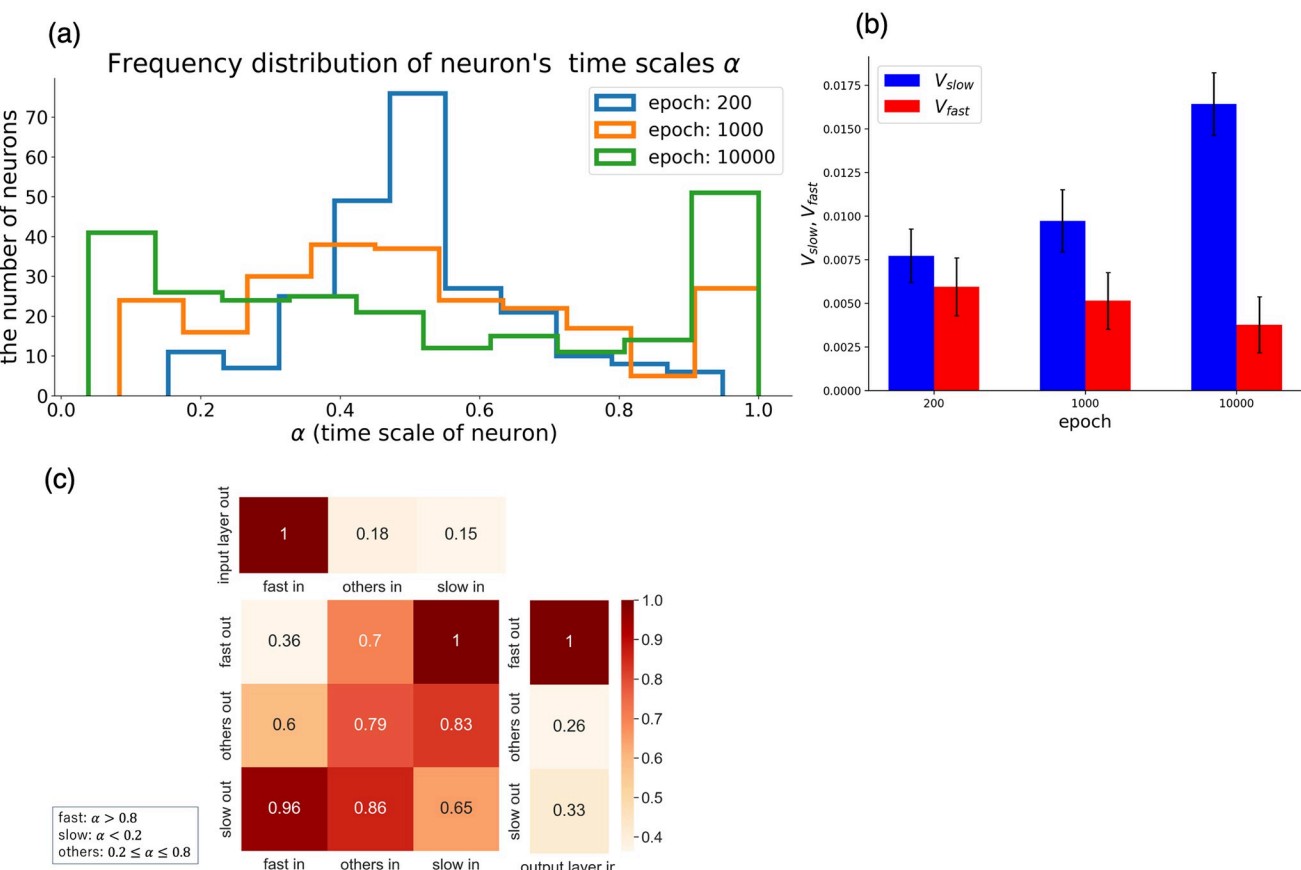

**Fig 9. RNN features obtained by learning when $\alpha$ is variable by learning.** (a) Frequency distribution of $\alpha$ for all neurons at 200, 1000, and 10000 learning epochs. At 10000 epochs, the learning process was complete. (b) Division of roles $V_{slow}$ and $V_{fast}$. See the "Representation of the prior" section for definitions of $V_{slow}$ and $V_{fast}$; $V_{slow}$ ($V_{fast}$) was computed for neurons with $\alpha < 0.2$ ($\alpha > 0.8$) respectively. (c) The average degree of RNN connections of 10000 epochs. Connections between input layer, recurrent neurons with $\alpha < 0.2$, $0.2 \leq \alpha \leq 0.8$, $\alpha > 0.8$, and output layer. Each was normalized so that the maximum value was 1.

The RNN could learn to approximate Bayesian inference using the prior(approximating the generator distribution) in all conditions we tested. However, the accuracy was higher in the modular RNN; further, the accuracy was significantly higher when the time scale of the sub-module was moderately slower than that of the main module. In addition, the increase in accuracy was pronounced against a rapidly varying input, for which it was necessary to generate a prior that changes quickly. To achieve such accuracy with a slow sub-module, the sub-module was found to specifically represent the prior, indicating role differentiation between the representation of the prior and the representation of the observed signal (likelihood). Of note, such functional differentiation is caused by differences in time scales. This result is consistent with experimental observations in the brain in which areas that code the prior and likelihood in Bayesian inference are different [18–20](However, caution is required as there is also an experimental report showing that the prior and likelihood are encoded in the same brain area [43]). Finally, it was shown that a modular structure with distinct time scales was spontaneously organized in the RNN by learning.

It is important to note that a relatively slow time scale of the neuron population encoding the prior is required, but the difference between fast and slow neurons should not be excessive.

If the time scale is too small, the accuracy is decreased (Fig 3), in which case the sub-module is not responsible for representing the prior (Fig 5). This is because prior construction requires a larger time span to address changes in external input for a neural network with such a slow time scale.

It has been suggested that the time scale of neurons slows down hierarchically from the area where the signal is directly applied to the area where information is proceed [21–23]. This hierarchical structure, combined with modularity [44], is believed to be relevant to information processing [44–46]. Our findings indicate that modular structures with two-level time scales could handle slowly changing inputs. Handling more complex environmental shifts might necessitate a more multi-layered modular structure with diverse time scales. With such a structure, Bayesian inference against complex temporal changes could be achieved by extrapolating the results of this study. Further research verifying this finding will elucidate the significance of hierarchical structuring in the brain. Notably, our simulations revealed that the distinction in time scales not only improves Bayesian inference accuracy but also spontaneously arises from learning processes. Considering these findings, a similar process may be expected in evolution [47].

The modular network with slow/fast time scales could integrate out noise and distinguish the average change in the inputs from fast noise. In fact, the network could effectively predict temporal changes in the input, even under rapidly changing conditions. The brain must adapt to time-varying, noisy inputs; hence, the performance of Bayesian inference by the network design reported herein is considered relevant to brain information processing.

We adopted a simple RNN and trained it using backpropagation. Backpropagation is often argued to be different from the learning algorithm implemented in the actual brain [48, 49], so care should be taken when generalizing our results. However, previous studies have suggested that neural networks trained by backpropagation can show similar behavior to that of the actual brain [38, 50–55]. For instance, by training neural networks by backpropagation, it is possible to produce a neural activity that displays the same behavior as place cells, which represents one's own spatial position [56]. It is generally considered that the learning scheme in the brain will not adopt backpropagation. Still, one may expect that neural networks and dynamics that achieve the requested task and Bayesian inference have a common structure, as long as the learning scheme is based on synaptic changes depending on on/off neural activity dynamics. Then, the present finding that neurons with slower time scales play a role in representing the prior will be relevant as a plausible explanation of how the brain actually behaves.

Unravelling the relationship between the structure of neural networks, neural dynamics, and the information processing performed by the brain is a primary goal in computational neuroscience [25–27, 57]. In this study, the relevance of modular structure and time scale difference in neural dynamics to the representation of the prior in Bayesian inference is demonstrated, as well as their formation by learning [58, 59], which will support ongoing research in the field.

## Supporting information

**S1 Fig. Trajectory of the internal state $\mathbf{x}_{sub}(t)$ of the sub-module when generator A $((\mu_A, \sigma_A^2) = (-0.5, 0.04))$ and generator B$((\mu_A, \sigma_A^2) = (0.5, 0.04))$ switch alternately.** Here the trajectories of the internal state $\mathbf{x}_{sub}(t)$ are plotted by the first and second principal components in S1 Fig for the cases in which generators A and B switch every 2-time steps and every 30-time steps. Generators A and B both have $\sigma_g = 0.04$. In the case of switching every 30-time steps, they were located in the region taken by the internal state when $\sigma_g$ was small. In the case of switching every 2-time steps, they were located in the region taken by the internal state when $\sigma_g$ was large. This occurred because the generators switched so rapidly that the RNN recognized that the signal was created by a generator with a large variance. This made it possible

to switch $y(t)$ quickly because the information of the observed signal $s$ was prioritized over the prior information when calculating the output $y$.
(EPS)

**S2 Fig.** Results of the RNN with $\alpha_m = \alpha_s = 0.1$: (a) Mean squared error between the optimal value $y_{opt}(t)$ and the output of RNN $y(t)$, plotted against the setting $(\alpha_m, \alpha_s) = (1, 1)$, $(1, 0.1)$, $(0.1, 0.1)$. (b) Division of roles for representing prior distribution. $V_s$, $V_m$ defined in the text Eqs (12) and (13) plotted for different values of $(\alpha_m, \alpha_s) = (1, 1)$, $(1, 0.1)$, $(0.1, 0.1)$ computed over 1000 samples of data. When $(\alpha_m, \alpha_s) = (0.1, 0.1)$, it resulted in $V_s < V_m$ and this indicates that the sub-module was unable to process prior-based information.
(EPS)

**S3 Fig.** Results of the different $N_s$, $N_m$: (a) Mean squared error between the optimal value $y_{opt}(t)$ and the output of RNN $y(t)$, plotted against the setting $(N_s, N_m) = (10, 240)$, $(50, 200)$, $(100, 150)$, $(150, 100)$, $(200, 50)$, $(240, 10)$. $\alpha_s$ is set to 0.1. (b) Division of roles for representing prior distribution. By fixing the sum $N_s + N_m$ to be constant at 250, we examined six configurations: $(N_s, N_m) = (10, 240)$, $(50, 200)$, $(100, 150)$, $(150, 100)$, $(200, 50)$, $(240, 10)$ while keeping $\alpha_s = 0.1$, $\alpha_m = 1$ and $p_t = 0.03$ fixed. Our results reveal that except for the cases $(N_s, N_m) = (10, 240)$, $(240, 10)$ efficient Bayesian inference, indicated by lower MSE values was observed for all other configurations(S3(a) Fig). The differences in MSE between the configurations $(N_s, N_m) = (50, 200)$, $(100, 150)$, $(150, 100)$, $(50, 200)$ were within the margin of error. Furthermore, the division of roles as measured by the variances ((Eqs (12) and (13))) between the sub-module and main-module was evident in all configurations except for $(N_s, N_m) = (10, 240)$(S3(b) Fig). As long as the number fraction is not too biased, the efficient Bayesian inference was achieved, with the division of roles. If the fraction of $N_s$ is too low, the variance for the slow module is larger, but the number of slow module is not sufficient to make appropriate Bayesian difference, whereas if it is too high the separation of variances does not follow. From these findings, it can be inferred that the results presented in this paper hold broadly, as long as neither of the modules is extremely undersized.
(EPS)

**S4 Fig. Extended examination of $\alpha_s$ in the range $0.16 - 0.06$: MSE between the optimal value $y_{opt}(t)$ and the output of RNN $y(t)$, plotted against the time scale $\alpha_s$.** Trained and tested the model with (a)$p_t = 0.03$ and (b)$p_t = 0.1$. In our present analysis, it was observed that if $\alpha_s$ is slow, accurate Bayesian inference is achievable. In Fig 3, MSE turned to be larger for $\alpha_s \lesssim 0.01$ or $\gtrsim 0.2$, and it was smaller around $\alpha_s \sim (0.05 \sim 0.2)$. Motivated by this, we investigated if detailed differences in $\alpha_s$ might pinpoint an optimal value, and if the outcomes would be influenced by variations in $p_t$. Here we change $\alpha_s$ values ranging from 0.16 to 0.06, as shown in S4 Fig, there was no significant differences within this range. Additionally, such insensitivity was observed irrespective of the differences in $p_t$.
(EPS)

**S5 Fig. Time scale $\alpha$ for different $p_t$: Frequency distribution of $\alpha$ for the model trained in (a) $p_t = 0.03$ setting, (b) $p_t = 0.1$ setting, and (c) $p_t = 0.3$ setting.** The peak on the smaller $\alpha$ side shifted to a larger value as $p_t$ increased. On the other hand, the peak on the larger $\alpha$ remained constant at 1.
(EPS)

## Acknowledgments

We thank Koji Hukushima and Yasushi Nagano for stimulating the discussion.

## Author Contributions

**Conceptualization:** Kohei Ichikawa, Kunihiko Kaneko.

**Data curation:** Kohei Ichikawa.

**Formal analysis:** Kohei Ichikawa.

**Investigation:** Kohei Ichikawa.

**Methodology:** Kohei Ichikawa.

**Supervision:** Kunihiko Kaneko.

**Visualization:** Kohei Ichikawa.

**Writing – original draft:** Kohei Ichikawa.

**Writing – review & editing:** Kohei Ichikawa, Kunihiko Kaneko.

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
