## [Decision Letter · Decision Letter 0]

7 Apr 2023

Dear student Ichikawa,

Thank you very much for submitting your manuscript "Bayesian inference is facilitated by modular neural networks with different time scales" for consideration at PLOS Computational Biology.

As with all papers reviewed by the journal, your manuscript was reviewed by members of the editorial board and by several independent reviewers. In light of the reviews (below this email), we would like to invite the resubmission of a significantly-revised version that takes into account the reviewers' comments.

Note especially that some of the reviewers highlighted issues with the clarity of the writing.

We cannot make any decision about publication until we have seen the revised manuscript and your response to the reviewers' comments. Your revised manuscript is also likely to be sent to reviewers for further evaluation.

Sincerely,

Ulrik R. Beierholm

Academic Editor

PLOS Computational Biology

Marieke van Vugt

Section Editor

PLOS Computational Biology

Reviewer's Responses to Questions

**Comments to the Authors:**

Reviewer #1: The authors present computational work showing that RNN can learn to represent prior and likelihood information in a close-to-optimal manner by obeying Bayes rule, and that this is accomplished by separated submodules that specialize to represent the prior with slow neurons and the likelihood with fast neurons. Overall, the paper is interesting and provide some novel results.

The introduction can be improved. It does not make justice to the interesting results that follow in the Results section. Also, in general the quality of English can be improved.

I wonder what would be the optimal architecture in the limit where p_t (the transition probability) goes to zero. I would assume that in this limit no modular architecture is needed, as the prior is fix and does not change over time (and it can be stored in the weights of the network). Could the author perform simulations of this case, and compare to the limit of p_t going to zero, as in Fig. 3b?

More details describing the learning method is needed. It is unclear how learning works when the prior distribution is changed over time with probability p_t. Is the prior relearn at each step? Or one uses all the data to train the network? For how long episodes? If yes, the network will naturally adapt to the time scale of the changes, but more discussion is needed.

The authors claim in the discussion that in the brain different subnetworks are in charge of encoding priors and likelihoods. I think that the authors are right that generally speaking slow, high level variables are encoding in higher brain areas than fast, low level variables. However, there are some exceptions. For instance, in the work by Mochol et al, Current Biology, 2021, it is shown that prefrontal cortex encodes both priors and likelihoods in the same brain area. Therefore, I would rewrite the discussion and being more cautious about this.

In Fig. 8, I would not use transparent colors to display the distributions.

Line 3: animal brain -> animal brains

Line 83: are the noise terms assumed to be independent

Line 88-89: ‘and’ in cursive

Line 208: unclear why y depends explicitly on s. I thought that it would depend only on x_m explicitly.

Reviewer #2: When performing Bayesian inference, the "prior" often changes abruptly, and for inference to be accurate these changes must be tracked. Here the authors investigate a modular network with both slow and fast timescales. They find an optimal timescale in the slow network. And, they also show that the timescale, and modular structure, can be learned, something that is very important.

Overall, this is an interesting paper that, I think, should be published. However, I got pretty lost in places, and the authors should clarify things before it's released to the public. The following comments should clarify my confusion (and also point out other issues, which were more minor).

1. Larger font on figures would be greatly appreciated -- some were _very_ hard to read (because I'm old ;)).

2. l 80, typo, I think: "= 0"  "= z".

3. Fig. 3: the y-axes should start at zero, to make it easy to see how big the effect is.

4. Fig. 3 (especially blue points in panel b): why not do more runs to decrease the error bars?

5. l 168: Fig. 2b should be Fig. 3b, I think.

6. l 195-232: Here I got very lost. You talk about setting mu_g and sigma_g for each module. But you can't do that, right? Instead, all you can do is determine how the input is generated. So the analysis doesn't make sense to me.

7. l 264: I got somewhat confused here as well. It seems to me that mu_p and sigma_p are internal values that are not directly accessible. If so, how can you find a transformation from x to mu_p (l 272)?

That said, the result of this analysis -- that alpha_s=0.1 uses more past information than alpha_s=1 -- is not surprising. But it doesn't explain why alpha_s=0.1 is better for rapid switches. Maybe that explanation was where I got lost?

Reviewer #3: # Paper synopsis

In the paper *Bayesian inference is facilitated by modular neural networks with different time scales*, Kohei Ichikawa and Kunihiko Kaneko explore how recurrent neural networks capture short and long timescales effects in a dynamic Bayesian inference problem. In particular, they study how a modular RNN architecture with different timescales for their respective neural activity dynamics can better predict a noisy stimulus than an RNN with a simpler architecture. The paper quantitatively assesses numerous features of the modular architecture, and overall demonstrates that the modular RNN learns to represent long-time scale, prior information in the slow timescale submodule, whereas the fast timescale module learns to focus on the immediate problem of identifying the noisy stimulus.

# Review and general recommendations

Overall I did like aspects of this paper, and I think the questions the authors raise about the role of timescales in inference are quite interesting. Moreover, the simulations they perform take some steps towards answering them. Nevertheless, the paper in its present form has fundamental issues:

(i) Overall the paper feels underdeveloped. Although particular explanations are written well enough, a lot of the text feels meandering, incomplete, and hard to follow. The figures also suffer from often conveying relatively little information, and with poor formatting and hard to read text. As such, this is a frustrating paper to review, because although I think some of the content is good, I feel like it's conveyed quite poorly. I feel the authors sent off this manuscript prematurely.

(ii) Similarly, the simulations presented in the paper are extremely limited both in breadth and quantity. As a neural network paper where the simulations are presumably not compute-intensive, I would like to see much broader testing of the hyper parameters of the network, from timescales, to number of neurons, to switching probabilities. Can the authors provide cases where the modularity of the network is *not* necessary? If the authors wish to support their claims with simulations rather than analysis, then the simulations must be to a much higher standard.

Overall, based on the current amount of content, it feels like a tightly written version of this paper could fit into 5 pages. I apologize that such a comment does not provide the authors with much guidance, but in its present form the paper feels too underdeveloped to identify exactly where to fill it in. These issues prevent me from recommending the paper for publication, and the authors will have to undertake major revisions to prepare it for publication.

# Local comments

(2023-04-07, 10:45:14 a.m.)

“when performing Bayesian inference by the brain.” (p. 1) redundant

“uncertainty” (p. 2) please cite Sokoloski in Neural Computation, 2017

“the” (p. 3) a

“the results to be discussed are 76 not altered as long as both the numbers are sufficient (say 100 vs. 50, 150 vs. 150 for 77 fast and slow neurons)” (p. 4) a bit more detail

“xmandxs” (p. 4) typo

“αmandαs” (p. 4) typo

“Then” (p. 5) this connecting word is confusing. Does this sentence follow from the previous?

“the ease of firing” (p. 5) tuning curve width?

“Therefore, training was performed to minimize the mean squared error (MSE) between the neural network output y(t) and the true value ytrue(t)” (p. 5) this is a unrealistic assumption and should be justified more.

“Hyperparameters” (p. 6) the amount of detail here is excessive. Focus on the key parameters

“pt” (p. 6) Why isn't this variable on the other side?

“Fig 4.” (p. 8) This figure only provides a weak demonstration of the point

“The network with αs = 1 could not follow rapid input changes, whereas that with αs = 0.1 could estimate the input prior effectively.” (p. 8) it would be nice to have some kind of explanation of how the alpha s timescale compared with the timescale of the switching. Is there an optimal alpha s that depends on switch speed?

“Fig 6” (p. 10) Labels and text are tool small in figures. Normalize text sizes across all figures.

“SI for the case” (p. 11) supplementary information? Please improve reference

“Effects of different time scales” (p. 11) This section seems far harder to read than it should be, given the simplicity of the point the authors are trying to make

“To examine the impact of αs differences on Bayesian inference accuracy in detail, we 258 considered how each model with αs = 1 and αs = 0.1 represents prior as a function of 259 the input signal.” (p. 11) At some point I would like to see key model hyperparameters thoroughly grid searched

“ak defined by Eq.14 is plotted against t, for the model with αs = 1 and αs = 0.1 using 3000 data points.” (p. 12) Figure text is unhelpful. It should be possible to have some understanding of the figure based on the caption

“Then,” (p. 12) You mean to say this is what this section is going to be about? This makes it sound like you've already done it

“normal RNN” (p. 12) reference earlier equations?

“In summary, a modular structure, shown in Fig.1(b), 315 emerged through learning alone.” (p. 12) The way this section is set up, it feels like it trivializes the earlier results of the paper. You could (and maybe should) simply start the paper with this section, show that 0.1 is optimal, and then analyze the results throughout the rest of the paper. I spent a lot of time in this paper trying to guess why the authors chose particular settings for various hyperparameters.

“Future research should investigate 342 how this optimal time scale depends on the time scale of environmental changes.” (p. 13) I think this research needs to be in this study

**Have the authors made all data and (if applicable) computational code underlying the findings in their manuscript fully available?**

Reviewer #1: Yes

Reviewer #2: Yes

Reviewer #3: Yes

PLOS authors have the option to publish the peer review history of their article (what does this mean?). If published, this will include your full peer review and any attached files.

Reviewer #1: No

Reviewer #2: No

Reviewer #3: **Yes: **Sacha Sokoloski
---

## [Decision Letter · Decision Letter 1]

29 Nov 2023

Dear student Ichikawa,

Thank you very much for submitting your manuscript "Bayesian inference is facilitated by modular neural networks with different time scales" for consideration at PLOS Computational Biology. As with all papers reviewed by the journal, your manuscript was reviewed by members of the editorial board and by several independent reviewers. The reviewers appreciated the attention to an important topic. Based on the reviews, we are likely to accept this manuscript for publication, providing that you modify the manuscript according to the review recommendations.

Sincerely,

Ulrik R. Beierholm

Academic Editor

PLOS Computational Biology

Marieke van Vugt

Section Editor

PLOS Computational Biology

Reviewer's Responses to Questions

**Comments to the Authors:**

Reviewer #1: Thanks for the improvements in the paper.

Reviewer #2: I still like this paper, and it's better, but it's also still confusing in places. Comments follow, more or less in order of appearance.

1. Typo in Eq. 2: should be N_m < i, not N_m \\le i.

2. How can you run with alpha=1? In that regime doesn't x diffuse off to infinity?

3. How often is u sampled? Every time step? Or only when things change?

4. p_t is introduced on line 221, but I don't think you ever told us what it is.

6. Lines 238-9, "When a(t) is close to 1, the model’s prior is closer to generator B, and when a(t) is close to -1, it is closer to prior A." Shouldn't it be 0 and 1, not 1 and -1?

7. Fig. 2 is a bit unsatisfying. What we would like to see is a plot of y versus s for different values of mu_g and sigma_g. Along with a summary statistic.

8. The mu_p explanation was confusing. As far as I can tell, you fit a matrix to mu_g = W_{mu_p} x, and then declared that mu_p = W_{mu_p} x? Is that correct? If so, you should say so; if not, I'm lost.

9. Lines 362-4: "Here, note that the findings that the modular structure with slow/fast time scales works better for Bayesian inference do not necessarily imply that the structure can be achievable just by learning." I believe you're just saying that it's not clear that alpha can be learned. If so, you should say that. If not, I'm lost.

Reviewer #3: This article is much improved, and the authors have addressed the bulk of my

concerns. Here are some additional comments to help further improve the manuscript.

p. 3, l. 44: "...corresponds to the higher layer in the brain." This statement

needs to be rephrased and made more precise.

p. 3, l. 59: "...the prior more appropriately than regular RNN." explain what

you mean by this.

p. 3, Eq. 1: Make sure to introduce all variables in the equation, even if you

can only fully define them later in the text.

p. 4, l. 153: You should state somewhere here exactly which parameters of the

neural network you're optimizing.

p. 10, l. 279: "...strongly reflects the difference in the prior distribution to

the difference in output, respectively." This sentence is unclear.

**Have the authors made all data and (if applicable) computational code underlying the findings in their manuscript fully available?**

Reviewer #1: Yes

Reviewer #2: Yes

Reviewer #3: Yes

PLOS authors have the option to publish the peer review history of their article (what does this mean?). If published, this will include your full peer review and any attached files.

Reviewer #1: No

Reviewer #2: No

Reviewer #3: No

Figure Files:

Data Requirements:

Reproducibility:

References:

---

## [Decision Letter · Decision Letter 2]

6 Feb 2024

Dear student Ichikawa,

We are pleased to inform you that your manuscript 'Bayesian inference is facilitated by modular neural networks with different time scales' has been provisionally accepted for publication in PLOS Computational Biology.

Best regards,

Thomas Serre

Section Editor

PLOS Computational Biology

Marieke van Vugt

Section Editor

PLOS Computational Biology

Reviewer's Responses to Questions

**Comments to the Authors:**

Reviewer #2: I'm happy with the revisions.

I do, however, have one more comment. The first few times I looked at Eq. 1, I thought that I was the identity matrix and alpha was a diagonal matrix. However, I just noticed that I and alpha are are vectors. That's fine, but if so you need to make it clear that you're using element-wise multiplication; e.g., alpha x is a vector whose i^th component is alpha_i x_i. Alternatively, you could make this standard and change I to the identity matrix and alpha to a diagonal matrix.

Either is fine with me, and I definitely don't need to see the paper again.

**Have the authors made all data and (if applicable) computational code underlying the findings in their manuscript fully available?**

Reviewer #2: Yes

PLOS authors have the option to publish the peer review history of their article (what does this mean?). If published, this will include your full peer review and any attached files.

Reviewer #2: No

---

## [Editor Report · Acceptance letter]

27 Feb 2024

PCOMPBIOL-D-23-00148R2 

Bayesian inference is facilitated by modular neural networks with different time scales

Dear Dr Ichikawa,

I am pleased to inform you that your manuscript has been formally accepted for publication in PLOS Computational Biology. Your manuscript is now with our production department and you will be notified of the publication date in due course.

With kind regards,

Zsofia Freund
